# Investigation of the Efficacy of Dithiothreitol and Glutathione on In Vitro Fertilization of Cryopreserved Large White Boar Semen

**DOI:** 10.3390/ani12091137

**Published:** 2022-04-28

**Authors:** Mahlatsana Ramaesela Ledwaba, Masindi Lottus Mphaphathi, Mamonene Angelinah Thema, Cyril Mpho Pilane, Tshimangadzo Lucky Nedambale

**Affiliations:** 1Agricultural Research Council, Animal Production, Germplasm Conservation and Reproductive Biotechnologies, Private Bag X2, Pretoria 0062, South Africa; mrledwaba5@gmail.com (M.R.L.); mamonenethema@gmail.com (M.A.T.); cyril@arc.agric.za (C.M.P.); nedambaletl@tut.ac.za (T.L.N.); 2Department of Animal Science, Tshwane University of Technology, Private Bag X680, Pretoria 0001, South Africa; 3Department of Animal, Wildlife and Grassland Science, University of the Free State, Bloemfontein 9300, South Africa

**Keywords:** dithiothreitol, glutathione, semen, in vitro fertilization

## Abstract

**Simple Summary:**

Boar sperm has proven to be extremely susceptible to cold shock and sensitive to peroxidative damage due to the high membrane content of polyunsaturated fatty acids. As a result, free radicals and oxidative stress can have a significant impact on the outcomes of in vitro fertilization (IVF). The objectives of this study were to evaluate the properties of sperm motility, viability, and morphology under induced oxidative stress; compare the antioxidant capacity of dithiothreitol (DTT) and glutathione (GSH) following the cryopreservation of Large White boar semen; investigate the ability of the cryopreserved Large White boar semen to fertilize the matured gilts oocytes under the same conditions; and to compare the efficacy of DTT and GSH antioxidants in improving the oocyte fertilization by the cryopreserved Large White boar semen. In this study, hydrogen peroxide (H_2_O_2_) was used as a source of oxidative stress (control). The treatment of boar semen with H_2_O_2_ reduced the percentage of sperm progressive motility (PM; 86.70 ± 10.24) after 3 h of incubation period (*p* < 0.05). The combination of GSH + DTT treatment reduced the percentage of sperm total motility (TM; 22.45 ± 11.14), PM (10.41 ± 4.59) and rapid motility (RAP; 9.20 ± 3.55) following thawing (*p* < 0.05). For IVF, fresh semen (11.84%) and a combination of DTT + GSH (14.10%) recorded a high percentage of zygotes with >2 pronucleus. However, the DTT treatment recorded a high percentage of zygotes with two pronuclei (19.76%).

**Abstract:**

The objectives of this study were to evaluate the properties of sperm motility and morphology under induced oxidative stress, compare the antioxidant capacity of dithiothreitol (DTT) and glutathione (GSH) following the cryopreservation of Large White boar semen, investigate the ability of cryopreserved Large White boar semen to fertilize the matured gilts oocytes and compare the efficacy of DTT and GSH antioxidants in improving the oocyte fertilization by cryopreserved Large White boar semen. The semen was collected from three Large White boars (ten ejaculates per boar) and transported (37 °C) to the laboratory. Semen freezing extenders were supplemented with 5 mM DTT, 5 mM GSH and a combination of 2.5 mM DTT + 2.5 mM GSH. A liquid nitrogen vapor method was used to freeze boar semen. Gilts’ ovaries were collected from the local abattoir and transported (37 °C) to the laboratory. The slicing method was used to retrieve the oocytes from the ovaries. Fresh semen and frozen-thawed semen were used for in vitro fertilization (IVF). For frozen-thawed semen, four treatments (control, 5 mM DTT, 5 mM GSH, and a combination of 2.5 mM DTT + 2.5 mM GSH) were used during IVF in order to evaluate the fertilizing ability of the antioxidants. The supplementation of 5 µM DTT to H_2_O_2_-treated semen significantly improved progressive motility (PM) by 14.82%. A combination of 2.5 mM DTT + 2.5 mM GSH treatment reduced percentage of sperm total motility (TM) and rapid motility (RAP) following thawing (*p* < 0.05). Fresh semen and a combination of 2.5 mM DTT + 2.5 mM GSH treatment recorded a higher percentage of zygotes with polyspermy (*p* < 0.05). The control treatment numerically recorded a high percentage of zygotes with 1 PN, while the 5 mM DTT treatment recorded a high percentage of zygotes with 2 PN.

## 1. Introduction

Cryopreservation is the most practical approach for the long-term storage of sperm in boars [1]. However, the cryopreservation and thawing processes result in cryodamage of frozen-thawed sperm quality, and fertilizing ability, as studied by Namula et al. [2]. It has been recognized that there is a reduced fertilization efficiency of boar sperm following freezing and thawing. Some hypotheses have been proposed to explain this phenomenon, and oxidative damage was considered to be one of the main factors [3,4]. Oxidative stress is a process where cellular components become modified by reactive oxygen species (ROS) that eventually lead to cell death. Boar sperm, specifically, are rich in membrane polyunsaturated fatty acids [5], which are a target for ROS. In sperm, the imbalance between high levels of ROS and low levels of antioxidants can lead to cells experiencing an oxidative-stressed environment, which could damage or cause cellular morphological abnormalities [6]. Antioxidants have been proven to have the ability to disrupt or slow down the lipid peroxidation reaction, thereby reducing oxidative stress and damage [6].

During cryopreservation, extenders are often required; hence, an improvement of semen cryopreservation technologies requires in-depth knowledge of the properties of the extender [7]. For this reason, extenders have been supplemented with various supplements, including antioxidants [8]. Antioxidants play a vital role in preserving sperm motility, integrity, metabolism, and function by protecting the sperm against oxidative damage [9]. Numerous studies have recently revealed that the supplementation of antioxidants to freezing extenders can neutralize ROS and improve post-thaw boar sperm function [10,11]. The addition of antioxidants such as glutathione (GSH) to the freezing and thawing extender might improve the quality and fertilizing ability of frozen-thawed boar sperm [12,13], as the addition of GSH helps to maintain sperm motility [14,15] and to protect the sperm against oxidative damage. Dithiothreitol (DTT) is known as an antioxidant that decreases the protamine disulfide bond [16]. It provides a protective effect against apoptosis and oxidative damage [17]. Other studies have indicated that DTT can prevent hydrogen peroxide (H_2_O_2_)-meditated loss of boar sperm quality [18,19]. The addition of DTT seems to improve bull and human sperm motility during preservation or in the frozen state. However, this was never tested on the boar semen.

It is important to utilize an antioxidant to improve oxidative damage during boar sperm cryopreservation. The in vitro fertilization (IVF) procedure is an important technology for the utilization of cryopreserved sperm in livestock gene banks, as studied by Kikuchi et al. [20]. It has been demonstrated that IVF is a basic technology that enables the production of embryos in a large amount, which can be used for embryonic development research. The reduced fertilization efficiency of boar sperm after freezing and thawing is well recognized. Therefore, finding low cytotoxic antioxidants at a suitable concentration is very important in improving the frozen-thawed boar sperm quality, hence improving their fertilizing ability following IVF. This study aims to investigate the ability of cryopreserved Large White boar semen to fertilize the matured gilts oocytes and to compare the efficacy of DTT and GSH antioxidants in improving the oocyte fertilization by cryopreserved Large White boar semen.

## 2. Materials and Methods

### 2.1. Materials

Three Large White boars (approximately 3 years old) were used for this study. These animals were raised and trained on the farm as semen donors for AI (artificial insemination) purposes. They were maintained under uniform feeding and housing conditions. The animals were fed once daily and water was provided ad libitum. The study was carried out for the duration of three months. Ovaries were collected from the prepubescent gilts of unknown breeds from the local abattoir. A completely randomized design was used for this study. Chemicals and reagents were purchased from Sigma Chemical Co. (St. Louis, MO, USA), and Whitehead Scientific (Pty) Ltd. (Cape Town, South Africa) unless otherwise stated. Empty straws and the freezing consumables were purchased from Embryo Plus^®^ and Lion Bridge, South Africa.

### 2.2. Boars Semen Collection

Semen was collected from three Large White boars of proven good fertility using a gloved-hand method. The sperm-rich fraction was collected using a thermos flask (500 mL semen collection, MS Schippers) containing warm water (37 °C) and a glass beaker covered with a gauze filter (non-sterile Sterilux^®^ ES, 10 cm × 10 cm) to separate the gel fraction from the sperm-rich fraction. After collection, the semen was transported to the laboratory for evaluation. Semen pH was measured using the pH meter (Oaklon, EW35614-30, Cole-Parmer, East Bunker Court, Vernon Hills, IL, USA). The SpermaCue (Minitube, Venture Court, Verona, WI, USA) was used to estimate the sperm concentration (×10^6^/mL). The semen volume was measured by using the 500 mL reusable media glass.

### 2.3. Experiment I

Experiment I covered the characterization of boar semen under induced oxidative stress, a total number of 30 ejaculates (10 ejaculates from each Large White boar) were collected from three Large White boars, twice a week for the duration of 10 weeks. The preliminary study was conducted in order to select a suitable concentration (1 µM, 2 µM, 3 µM, 4 µM, and 5 µM) for H_2_O_2_, DTT, and GSH for liquid preservation.

#### Characterization of Large White Boar’s Semen under Induced Oxidative Stress

The hydrogen peroxide (H_2_O_2_) was used as a source of oxidative stress (control). For the semen treatment, 10 mM/mL of H_2_O_2_ stock solution was prepared in Medium 199 (M199) and kept at 5 °C until use. Semen from three Large White boars was pooled (in order to eliminate the individual differences), divided into six parts, and treated with pre-warmed H_2_O_2_ stock supplemented with 5 µM/mL of GSH and 5 µM/mL of DTT to make six treatments (control, 5 µM H_2_O_2_, 5 µM DTT, 5 µM GSH, 5 µM H_2_O_2_ + 5 µM DTT, 5 µM H_2_O_2_ + 5 µM GSH), then incubated at 30 °C for 3 h in a humified 5% CO_2_ and 95% atmospheric air incubator. After 3 h of incubation, a drop of 5 µL of semen was placed on a pre-warmed microscope slide (Labchem Pty Ltd., Cape Town, South Africa) and covered with a coverslip (Labchem Pty Ltd., Cape Town, South Africa), then examined under a microscope using a Sperm Class Analyser (Microptic S.L., Barcelona, Spain). The sperm motility and velocity traits measured (Table 1) include the sperm total motility (TM%), progressive motility (PM%), non-progressive motility (NPM%), static (STC%), rapid (RAP%), medium (MED%), slow (SLW%), curvilinear (VCL µm/s), straight-line (VSL µm/s), average path velocity (VAP µm/s), linearity (LIN%), straightness (STR%), wobble (WOB%), the amplitude of lateral head displacement (ALH µm/s), beat cross frequency (BCF Hz) and hyperactivity (HPA%). The sperm viability and morphology traits were evaluated using the Eosin/ Nigrosin (University of PretoriaPretoria, South Africa) staining. Briefly, a drop of 7 µL of the semen was added to the 20 µL of Eosin/Nigrosin staining. A drop of 5 μL mixed sample was placed on the end of the microscope slide and smeared to the other end of the microscope slide. Thereafter, the slide was left to dry off at room temperature for 5–10 min before evaluation. A fluorescent microscope (Olympus Corporation BX 51FT, Tokyo, Japan) was used at 100× magnification to count 200 sperm per stained slide. For this analysis, sperm viability (live and dead) was recorded and sperm abnormalities (live sperms with head defects, live sperms with tail defects, live sperms with midpiece defects, and live sperms with droplets defects) were recorded.

### 2.4. Experiment II

Experiment II covered the boar semen cryopreservation and evaluation; a total number of 30 ejaculates (10 ejaculates from each Large White boar) were collected from three Large White boars twice a week for the duration of 10 weeks. The preliminary study was conducted in order to select a suitable concentration (2.5 mM, 5 mM, 7.5 mM, and 10 mM) for DTT and GSH for the cryopreservation of Large White boar semen. The concentration was also supported by the literature review.

#### 2.4.1. Cryopreservation of Large White Boar’s Semen

In brief, following the semen analysis, the semen was transferred into 50 mL centrifuge tubes and diluted with Beltsville Thawing Solution (BTS) at a ratio of 1:1 *v*/*v*. Diluted semen was equilibrated at 17 °C for 2 h and later centrifuged at 800× *g* for 10 min at 15 °C. Following centrifugation, the supernatant was discarded and the sperm pellet was re-suspended with chicken egg yolk (20% egg yolk and 80% BTS) freezing extender at a ratio of 1:1 and placed back for cooling at 5 °C for 1 h 30 min. After cooling, the semen was divided into four parts to make four treatments (control, 5 mM GSH, 5 mM DTT, and a combination of 2.5 mM GSH + 2.5 mM DTT). Thereafter, semen samples were supplemented with four different chicken egg yolk freezing extenders, as follows: control (20% egg yolk, 3% glycerol and 77% BTS), 5 mM DTT (20% egg yolk, 3% glycerol, 76.5% BTS and 0.5% DTT), 5 mM GSH (20% egg yolk, 3% glycerol, 76.5% BTS and 0.5% GSH), and 2.5 mM GSH + 2.5 mM DTT (20% egg yolk, 3% glycerol, 76.5% BTS, 0.25% GSH and 0.25% DTT). After dilution, semen was loaded into 0.25 mL polyvinyl chloride straws. The semen straws were placed in contact with liquid nitrogen vapor about 3 cm above the liquid nitrogen level for 20 min in an expandable polystyrene box and later plunged directly into the liquid nitrogen tank (−196 °C) until thawing.

#### 2.4.2. Boars Semen Thawing and Evaluation

After two or three days of freezing, ten straws from each treatment were thawed for 10 s in the air and 1 min in the circular water bath at 37 °C. The straws were dried and cut using the sterilized scissor on both of the sealed ends and the semen was transferred into the Eppendorf tube. Immediately after thawing, the semen was evaluated for standard sperm traits, i.e., sperm motility, morphology, and acrosome integrity traits.

#### 2.4.3. Evaluation of Sperm Acrosome Integrity

For acrosome integrity, DAPI staining (Whitehead Scientific, South Africa) was used to evaluate the sperm acrosome integrity. Briefly, 200 μL of semen was diluted with pre-warmed 200 μL of Dulbecco’s phosphate-buffered saline into the centrifuge tube (Whitehead Scientific, South Africa). The samples were centrifuged at 800 g for 2 min. Following centrifugation, the supernatant was discarded and the semen pellet was re-suspended in 10 μL of DAPI staining, and then incubated at 37 °C for 10 min. After 10 min, a drop of 5 μL mixed sample was placed on the end of the microscope slide and smeared on the other end of the microscope slide. Thereafter, the slide was left to dry off at room temperature for 5 to 10 min. Acrosome integrity was assessed using DAPI staining under the fluorescent microscope at 100× magnification. For this analysis, sperms were classified as intact acrosome and non-reactant acrosome

### 2.5. Experiment III

Experiment III covered the evaluation of the ability of the cryopreserved Large White boar semen to fertilize the matured oocytes. A total of 250 ovaries (50 ovaries per treatment) were collected 5 days a week for the duration of 5 weeks. For this experiment, four treatments (fresh semen, control, 5 mM DTT, 5 mM GSH, and a combination of 2.5 mM DTT and 2.5 mM GSH) from the cryopreserved semen were used in order to evaluate their ability to fertilize matured oocytes.

#### 2.5.1. Ovaries Collection

Ovaries were collected from the slaughtered prepubescent gilts at the local abattoir. The collected ovaries were then placed inside the thermos flask and covered with 0.9% NaCl saline solution at 38 °C and transported from the abattoir to the laboratory within an hour. Upon arrival at the laboratory, the ovaries were washed with 0.9% saline water to remove excess blood; thereafter, they were sprayed with 70% alcohol for further prevention of contamination. The ovaries were transferred into a dish containing saline water and placed in a water bath at 38 °C.

The ovaries were cut free of any tissues on their surfaces, placed in a searching petri dish containing 2 mL mDPBS, and supplemented with 1% polyvinyl alcohol and 10% antibiotic. The ovaries were sliced with a sharp sterile blade. The ovaries were held using forceps over the dish to ensure easy slicing. The remaining fluid inside the petri dish was transferred into a 50 mL tube containing 5 mL of mDPBS. The supernatant was then gently removed using a Pasteur pipette without disturbing the pellet. The pellet was evaluated with the aid of a stereo microscope in order to search for immature oocytes (Figure 1A).

#### 2.5.2. In Vitro Maturation of Oocytes

The base maturation medium consisted of North California States University 23 (NCSU 23) supplemented with 10 ng/mL of follicle stimulating hormone, 10 ng/mL of luteinizing hormone and 10% porcine follicular fluid (pFF) recovered from prepubertal follicles. The fertilization medium consisted of modified Tris buffered medium (mTBM) containing 113.1 mM sodium chloride, 3 mM potassium chloride, 7.5 mM calcium chloride dihydrate, 20 mM Tris, 11 mM glucose, 5 mM sodiumpyruvate, 1 mM caffeine and 0.1% bovine serum albumin (BSA). For the staining of the oocytes, 25 mg (0.025 g) of Hoechst 33342 (Sigma B226) was prepared in 2.5 mL of pure water to make stock A. The concentration of the solution was 10 mg/mL. On the day of use, 10 μL of stock A was diluted in 10 mL of Dulbecco’s phosphate buffered saline (DPBS) supplemented with 20% (2 mL) glycerol to make stock B (8 mL of DPBS and 2 mL of glycerol). The final concentration for stock B was 10 μg/mL.

A total of six dishes (Falcon 1008) were prepared where three contained 3 mL of M199 + 10% fetal bovine serum (FBS) each and the other three contained 3 mL of mDPBS + 10% BSA. The six dishes were placed in the incubator at 38.5 °C prior to the ovaries collection. The oocytes were retrieved from the ovaries using the slicing method and washed three times in mDPBS + 10% BSA and washed again three times in M199 + 10% FBS. The oocytes were searched under the stereomicroscope (Olympus BX71, Taipei, Taiwan, which was placed under the lamina floor to ensure sterility. Only grade A (oocytes with three layers of cumulus cells) and B (oocytes with at least two layers of cumulus cells) oocytes were used for in vitro maturation (IVM). A total of 500 µL of NCSU-23 from the prepared medium was placed into sterile four-well dishes and covered with 250 µL of sterile mineral oil, to prevent evaporation. The dishes were placed inside the incubator at 38.5 °C prior to the ovaries collection. The oocytes were placed in a four-well dish containing pre-warmed IVM media and incubated in a Thermo^®^ CO_2_ incubator at 38.5 °C for 44 h. After 44 h of incubation, matured oocytes (Figure 1B) were identified by the presence of expanded cumulus cells and were used for IVF using fresh and frozen-thawed Large White semen.

#### 2.5.3. In Vitro Fertilization of Matured Oocytes and Thawing of Semen

Only cumulus-oocyte complexes were used for IVF. Frozen semen straws were thawed for 10 s in the air and 1 min in the thawing unit containing warm water at 37 °C. The straws were dried and cut using the sterilized scissor on both of the sealed ends and the semen was transferred into the 15 mL Eppendorf tube. Immediately after thawing, a drop of 5 µL of the semen was placed on the warmed microscope slide and covered with a coverslip. The Sperm Class Analyser^®^ (Microptic S.L., Barcelona, Spain) was used to evaluate sperm motility before fertilization. Fresh semen was collected one day before IVF, diluted with BTS, then equilibrated overnight at 17 °C. Sperm motility was also evaluated prior to IVF.

For sperm capacitation, fresh and thawed semen was diluted with 6 mL of pre-warmed (37 °C) IVF media. The mixture was centrifuged at 2000 rpm for 2 min at 37 °C. Following centrifugation, the pellet was formed at the bottom of the tube, and the top part was removed carefully using the sterile Pasteur pipette without disturbing the sperm pellet. The same level of IVF sperm wash was added to the pellet and the mixture was centrifuged for the second time at 2000 rpm for 2 min at 37 °C. After centrifugation, the supernatant was removed, leaving the pellet at the bottom of the tube. The sperm pellet was diluted with fertilization medium depending on the number of drops having oocytes and the concentration of the sperm.

Prior to IVF, five wash drops (100 μL) of mTBM supplemented with 1 mM caffeine + 0.1% BSA were made in 1008 falcon^®^ Petri dishes and covered with 3 mL of mineral oil to prevent evaporation. Another 2 drops of 50 μL of the same medium were also created in similar dishes; these drops were used as IVF drops. Firstly, matured oocytes were washed 5 times in pre-warmed IVF media covered with 3 mL of mineral oil. After washing, the oocytes were randomly distributed into the Petri dish containing 50 μL of the IVF media covered with 3 mL of mineral oil. A total of 50 μL of capacitated diluted sperm was added to each 50 μL drop containing oocytes. The IVF dishes were co-incubated at 38.5 °C in a moist atmosphere of 5% CO_2_ the air for 6 h.

#### 2.5.4. Assessment of the Pronucleus following In Vitro Fertilization of Porcine Oocytes

Fertilized oocytes were denuded by vortexing them for 3 min in pre-warmed M199 + 10% FBS solution. The oocytes were then transferred in a dish contacting M199 + 10% FBS. For staining of the oocytes, 25 mg (0.025 g) of Hoechst 33342 (Sigma B226, St. Louis, MO, USA) was used. Oocytes were transferred to the glass slide, then, 4 small drops of lubricant (Vaseline) were made around the zygote drop. A minimal volume (2–10 μL) of stock B solution was added to the zygote drop. The coverslip was placed over the slide and gently squeezed until it touched the zygote drop. Colorless nail polish was used to seal the ends of the coverslip. The slide was allowed to dry for 2 h in a dark compartment before counting the zygotes with the aid of an inverted microscope. The zygotes were evaluated for a total fertilization rate at 0 pronucleus (0 PN), 1 PN, 2 PN, and >2 PN.

### 2.6. Statistical Analysis

Data were analyzed using the Generalized Linear Model procedure. Treatment means were separated using the Fisher’s protected *t*-test, with the least significant difference at a 0.05 level of significance. Pearson’s correlation coefficients were calculated to test the relationships between the motility, velocity, and fertilization rate in the presence or absence of antioxidants. Percentage data are presented as mean ± standard deviation (Mean ± SD) values.

## 3. Results

The macroscopic evaluations of the Large White boar’s results are set out in Figure 2. The Large White boars’ semen volume ranged from 96 to 111 mL, with an average of 104.33 mL. The semen pH ranged from 7.46 to 7.56 with an average of 7.51. The sperm concentration ranged from 176.6 to 215 × 10^6^/mL with an average of 199.5 × 10^6^/mL. There was no significant difference found in semen volume, pH, and sperm concentration between the three boars. Raw sperm TM (97.93 ± 2.14) and RAP (40.63 ± 14.53) were recorded (Table 2). Fresh semen obtained from Large White boars exhibited 92.80% of viable sperm morphology (Table 3).

The properties of sperm motility and velocity traits following in vitro liquid preservation under the induced oxidative stress, which follows incubation for 3 h, are evaluated (Table 4). The sperm TM among treatments ranged from 86.70 to 93.32%. The control treatment recorded a high percentage of sperm PM (71.28 ± 17.86) and RAP (64.92 ± 21.60) after 3 h of the incubation period, as compared to 5 µM H_2_O_2_ (49.88 ± 16.81; 40.76 ± 19.69, respectively) and 5 µM DTT (54.14 ± 12.63; 43.48 ± 16.27, respectively) treated semen, respectively (*p* < 0.05). The addition of 5 µM H_2_O_2_ to the semen reduced the sperm PM (86.70 ± 10.24) after 3 h of the incubation period (*p* < 0.05). Interestingly, the supplementation of 5 µM DTT (64.70 ± 14.29) to 5 µM of H_2_O_2_-treated semen significantly improved the sperm PM by 14.82%, while a 5 µM GSH (59.31 ± 14.32) treatment non-significantly improved the sperm PM by 9.43% after 3 h of the incubation period. Furthermore, the supplementation of 5 µM DTT (54.44 ± 20.38) to 5 µM H_2_O_2_ (40.76 ± 19.69)-treated semen numerically improved sperm moving at a rapid rate by 13.68% after 3 h of the incubation period. The sperm velocity traits improved by 5 µM DTT including the VCL and VAP, though it revealed no significant difference. However, the supplementation of 5 µM GSH (31.76 ± 10.32) to 5 µM of H_2_O_2_ (24.61 ± 5.16)-treated semen improved (VSL) by 7.15% after 3 h of the incubation period (*p* < 0.05). No significant difference was recorded in sperm TM, WOB, BCF, and HPA among the treatments.

Sperm morphology of above 70% was recorded from Large White boars’ fresh semen (Table 5). The 5 µM DTT alone (85.10 ± 6.51) and control (86.00 ± 4.22) treatments had the highest viable sperm morphology after 3 h of the incubation period (*p* < 0.05). The percentage of live sperm with head and proximal droplet defects differed significantly among the treatments. Supplementation of 5 µM DTT to 5 µM H_2_O_2_-treated semen increased live sperm with head and proximal droplets defects (1.10 ± 1.0 and 1.40 ± 1.07, respectively), as compared to 5 µM H_2_O_2_ (0.30 ± 0.48; 0.50 ± 0.53)-treated semen (*p* < 0.05). There was no significant difference in live sperm with tail and distal droplets defects after 3 h of the incubation period among the treatments.

The average sperm TM for frozen-thawed semen was 27.23% for all treatments (Table 6). There was no significant difference recorded in sperm NPM, MED, VCL, VSL, VAP, LIN, STR, WOB, ALH, BCF, and HPA among treatments. The supplementation of 5 mM DTT (28.33 ± 10.66) and 5 mM GSH (26.04 ± 9.41) to freezing extenders maintained sperm TM following thawing and was significant to the control treatment (32.09 ± 6.66). The combination of 2.5 mM DTT + 2.5 mM GSH treatment reduced sperm TM, PM and RAP (22.45 ± 11.14; 10.41 ± 4.59; 9.20 ± 3.55, respectively) following the thawing of frozen semen straws as compared to the control treatment (32.09 ± 6.66; 19.13 ± 6.96; 17.04 ± 6.55), respectively (*p* < 0.05). Interestingly, the supplementation of 2.5 mM DTT + 2.5 mM GSH (123.01 ± 24.79) treatment into the freezing extender has numerically improved sperm curvilinear velocity by 11.38 µm/s, as compared to the control treatment (111.63 ± 18.29).

The average live sperm of frozen-thawed semen was 39.71% for all treatments (Table 7). The combination of 2.5 mM DTT + 2.5 mM GSH recorded the least viable sperm morphology (21.67 ± 6.91) following the thawing, as compared to the control (34.90 ± 6.51), with 5 mM DTT (29.80 ± 5.20) and 5 mM GSH (29.40 ± 6.38) treatments (*p* < 0.05). Dithiothreitol (1.70 ± 0.82) and 2.5 mM DTT + 2.5 mM GSH (1.70 ± 0.67) treatments numerically increased live sperm with tail defects. The supplementation of 5 mM DTT (0.50 ± 0.53) to freezing extenders numerically increased live sperm with proximal defects, while supplementation of 5 mM GSH (0.50 ± 0.53) to the freezing extender also increased the live sperm with head defects. There was no recorded significant difference in sperm acrosome integrity and live sperm with abnormalities for all treatments.

The results of the effect of antioxidants on sperm fertilization ability following the IVM of gilts oocytes in vitro are presented in Table 8. The total fertilization rate ranged from 31.94 to 48.72% (*p* < 0.05) among the treatments. No significant difference was recorded on zygotes with 1 PN, 2PN, total number fertilization rate, and total non-fertilization rate for all treatments. The fresh semen (11.84 ± 9.47) and the combination of 2.5 mM DTT + 2.5 mM GSH (14.10 ± 10.49) recorded a high percentage of zygotes with >2 PN as compared to the 5 mM GSH (0.86 ± 1.92) and 5 mM DTT treatments (2.22 ± 4.96), respectively (*p* < 0.05). The 2.5 mM GSH treatment recorded the least percentage of fertilization rate (31.94 ± 8.65), zygotes with 1 PN (18.32 ± 3.92), 2 PN (11.18 ± 5.06) and >2 PN (0.86 ± 1.92), as compared to all the treatments (*p* > 0.05). The control treatment (28.1%) recorded a high percentage of zygotes with 1 PN (Figure 3), while the 5 mM DTT treatment (19.8%) recorded a high percentage of zygotes with 2 PN (Figure 4), (*p* > 0.05).

The Pearson’s correlation coefficients between sperm parameter results, evaluated by the computer-assisted sperm analysis and fertilization rate of gilts oocytes, are set out in Table 9. The coefficients were lower to higher with negative or positive correlation values ranging from r = −0.09 to 0.97. There was a positive relationship (r = 0.28) between sperm rapid motility and zygotes with <2 PN (Figure 5). Surprisingly, the sperm total motility exhibited no correlation between fertilization rate (r = −0.06) and zygotes with 1 PN.

Figure 3, Figure 4 and Figure 5 represent the results of the in vitro fertilization of gilts oocytes. Meanwhile, Figure 3 represents a 1 PN, which indicates that fertilization occurred but only one gamete produced a pronuclear structure; however, the DNA from one gamete (the sperm) is missing and only 1 PN (female pronuclei) is present. Therefore, the average for the zygote with 1 PN for all treatments was 22.9% (Table 7). Figure 4 represents normal fertilization, whereby the zygote had 2 PN (female and male pronuclei). This figure indicates that only one sperm penetrated the matured oocyte and was able to fertilize the matured oocyte. Therefore, the fertilization was successful and the zygote can reach the in vitro culture stage and develop into an embryo. The average percentage for zygotes with normal fertilization (2 PN) was 13.22% for all the treatments (Table 7). Figure 5 represents polyspermy, whereby more than one sperm fertilized a matured oocyte. Even though fertilization occurred, the survival of the zygote to the embryo stage is low, as there is an uneven number of chromosomes due to the extra pronuclei. Therefore, the zygote with three pronucleus states tends to develop into poorer-quality embryos. The average percentage for polyspermy (>2 PN) for all the treatments was 7.21% (Table 7).

## 4. Discussion

The present study was conducted to measure the boar sperm kinematic, morphological, and acrosome integrity traits in an attempt to compare the antioxidant (glutathione and dithiothreitol) capacity in Large White boar semen under induced oxidative stress and following cryopreservation. Our finding demonstrated that the average semen volume ejaculated from the Large White boar was 104.33 mL. The Large White boars semen pH ranged from 7.46 to 7.56 with an average of 7.51 in the present study. This is in agreement with Johnson et al. [24], who recorded a semen pH that varied between 7.2 and 7.5. The semen pH is in relation to sperm motility and metabolism. A semen pH value that is outside of the normal range is detrimental to the sperm and can affect their ability to penetrate the oocyte. Moreover, if the semen pH is too acidic or alkaline, it can affect the sperm quality, which will result in subfertility or infertility. The sperm concentration of Large White boars ranged from 176 to 215 × 10^6^/mL with an average of 199.50 × 10^6^/mL. In the present study, there was no significant difference recorded in the sperm concentration among the three boars.

Hydrogen peroxide is described as one of the experimental sources of free radicals that induce oxidative stress, as studied by Pilane et al. [25]. As boar sperm have a high membrane polyunsaturated fatty acid content, they are extremely susceptible to cold shock and sensitive to peroxidative damage, as studied by Pilane et al. [26]. Therefore, the susceptibility of boar sperm to oxidative stress was mimicked by using H_2_O_2_. Glutathione and dithiothreitol were introduced as the two antioxidants that inhibit oxidation. In the present study, an overall sperm TM was recorded. The results from the present study indicate that supplementation of 5 µM DTT to 5 µM H_2_O_2_-treated sperm did not have an effect on the sperm TM percentage. The treatment of 5 µM DTT to 5 µM H_2_O_2_ treatment has been shown to protect the sperm from lipid peroxidation and also maintain the motility of the sperm. A similar study was conducted by Pilane et al. [26] and recorded sperm TM for 5 µM H_2_O_2_ (29.1%) and 5 µM H_2_O_2_ + 5 µM DTT (29.1%), which were lower compared to the present study. No significant difference was recorded between the 5 µM H_2_O_2_ + 5 µM DTT and 5 µM H_2_O_2_ + 5 µM GSH, as compared with the 5 µM H_2_O_2_ regarding the sperm TM, PM, and RAP traits. These findings indicated that both the DTT and GSH significantly performed equally in preventing the sperm from oxidation stress. The present study demonstrated no effect on the sperm traits, namely: VCL, VAP, LIN, STR, WOB, ALH, BCF, and HPA between 5 µM H_2_O_2_, 5 µM H_2_O_2_ + 5 µM DTT, and 5 µM H_2_O_2_ + 5 µM GSH. The findings in the present study demonstrate that supplementation of 5 µM DTT and 5 µM GSH to H_2_O_2_-treated semen can neutralize and reduce the effect of oxidative stress on sperm motility and viability traits. An overall normal live sperm percentage was recorded. The 5 µM DTT and control treatments recorded the highest viable sperm morphology. These results demonstrated that the supplementation of 5 µM DTT and 5 µM GSH to H_2_O_2_-treated semen has managed to maintain the viability of the sperm and mimic the chances of further damage to the sperm due to ROS. However, the supplementation of 5 µM DTT to H_2_O_2_ has been demonstrated to have significantly increased live sperm with proximal defects. Even though the supplementation of DTT to semen can improve sperm viability, it is detrimental to sperm abnormalities. The results from the present study demonstrate that the DTT treatment alone improves the sperm viability of Large White boars and also suppresses sperm abnormalities. However, when the 5 µM DTT is supplemented into the H_2_O_2_-treated semen, it can only stabilize the sperm viability, and instead results in increased sperm abnormalities.

In the present study, the overall sperm TM following thawing was recorded. With the supplementation of 2.5 mM GSH + 2.5 mM DTT, the freezing extender reduced the sperm motility traits following thawing. These findings demonstrate that the combination of 2.5 mM DTT and 5 mM GSH in the semen is detrimental, as it cannot protect the semen from cryo-injury during freezing. The sperm TM of the control treatment was in contrast with the sperm TM of the control treatment observed by Gadea et al. [13]. Kaeoket et al. [27] recorded the sperm TM for a 5 mM GSH treatment when they were comparing the traditional nitrogen method (23.2%) and the control rate freezer method (24.6%) with the semen from Landrace, Duroc, Large White, and Pietrain breeds. Their results were comparable to the sperm TM for a 5 mM GSH treatment obtained from the present study. The results in the present study revealed that there was no significant difference in the sperm TM, PM, and RAP between the control, 5 mM DTT, and 5 mM GSH treatment, although numerically, there is a moderate difference between the three treatments. This indicates that semen that is cryopreserved without the antioxidants can survive the cryo-injury and maintain moderate sperm motility. The results also revealed that during freezing, the sperm experienced high osmotic and physical damage due to the formation of intracellular ice crystals, later it deteriorated sperm functions. Supplementation of both Dithiothreitol and Glutathione to freezing extenders was able to maintain the sperm motility, though there was no improvement. Therefore, supplementation of 5 mM DTT, 5 mM GSH, or a combination of 2.5 mM GSH + 2.5 mM DTT did not have an effect on the sperm velocity traits, even though they had an influence on the sperm TM, PM, and RAP traits.

In the present study, the supplementation of 2.5 mM GSH + 2.5 mM DTT to the freezing extender reduced sperm viability following thawing. The 5 mM DTT and 2.5 mM GSH + 2.5 mM DTT treatments numerically increased live sperm with tail defects. This indicates that the concentration of 2.5 mM DTT in the combined treatment (2.5 mM GSH + 2.5 mM DTT) has more impact on the increase in the sperm tail abnormalities following thawing. There was no recorded significant difference in sperm acrosome integrity following thawing. Even though no significant difference was recorded in live sperm with abnormalities among treatments, the supplementation of 5 mM DTT to the freezing extender has proven to have a harmful effect on the sperm tail, and proximal and distal defects. The present study demonstrated that the supplementation of antioxidants to the freezing extenders during the cryopreservation of boar semen can protect the sperm from cryo-injury and also improve the reduction of sperm viability and acrosome integrity following thawing. Therefore, the use of antioxidants during the cryopreservation of boar semen may prove to maintain sperm motility, morphology, and acrosome integrity traits.

Sperm fertility is an important factor to be considered during in vitro fertilization. The presence of two pronuclei is the first sign of successful fertilization, as observed during IVF. In this study, fresh and frozen-thawed semen was used for the IVF of porcine oocytes in order to determine the fertilizing ability of frozen-thawed semen supplemented with antioxidants. The 5 mM GSH treatment had the least total fertilization rate following IVF. However, the polyspermy (>2 PN) rates up to 0.86% were obtained, which indicated that there is a high chance of good quality embryos following IVC. The fresh semen treatment recorded 22.50% of zygotes with 1 PN. This process occurs when a defective sperm penetrates into a matured oocyte, signaling the oocyte to form its pronucleus. Meanwhile, the defective sperm is incapable of forming its own pronucleus. These results were supported by the results from a previous study [28], whereby fresh semen recorded 24.40% of zygotes with 1 PN, which also revealed that with only one functional PN (one-half of the chromosomes), the production of an embryo is incomplete. The control treatment recorded 28.12% of zygotes with 1 PN.

It has been discovered by Reichman et al. [29] that two-pronuclear zygotes that transitioned through 1 PN or 3 PN positions are likely to develop into poorer-quality embryos than ones that remain at 2 PN throughout the development, and might be significant in embryo selection in IVF. When evaluating the pronucleus formation following IVF, higher zygotes with >2 PN were recorded in 2.5 mM DTT + 2.5 mM GSH (14.10%), fresh semen (10.02%) and control treatment (7.04%). The results for the control treatment were in line with the results found in a previous study conducted [24], whereby frozen-thawed ejaculate semen recorded 12.30% of zygotes with >2 PN. Fresh semen recorded 10.02% of zygotes with polyspermy. The results by Casillas et al. [30] supported the results from the present study, which demonstrated that fresh semen grades in zygotes with polyspermy, later affected the quality of embryos following IVF. The presence of polyspermy in zygotes is a major problem, as it causes abnormal chromosome numbers in embryos [31]. Though most embryos with chromosomal abnormality have failed to develop into a fetus [32,33], some polyspermic zygotes have the ability to develop into diploid embryos [34] and even into live piglets [35]. The polyspermy is so dominant in porcine IVF systems due to the method of washing sperm by centrifugation in fertilization media containing capacitating factors (calcium and bicarbonate), and supplementation of capacitating molecules (BSA and caffeine) [36]. This procedure induces the increased and rapid contact of capacitated sperm with the female gametes. Therefore, the intracytoplasmic sperm injection should be considered instead of traditional in vitro fertilization, as it can reduce or eliminate the possibility of polyspermy entering a matured oocyte during IVF. It was found that the percentage of sperm moving rapidly and progressively had a negative correlation with two pronucleus polyspermic zygotes. Moreover, the sperm total motility was demonstrated to have a negative correlation with the fertilization rate. However, the polyspermic zygotes proved to have a positive correlation with sperm rapid motility and total motility.

In the present study, two different antioxidants were supplemented in freezing mediums in order to reduce the cryo-damage that occurs during the cryopreservation of semen from Large White boars. During this process, the semen was also used for IVF in order to observe the ability of semen to fertilize the matured oocytes. Although both treatments were able to fertilize the matured oocytes and protect semen during freezing, the aim was to compare their efficacy during liquid preservation, cryopreservation, and IVF in order to conclude on the antioxidants that were more efficacious. Both antioxidants have proven to be efficient in maintaining semen quality and fertility during liquid preservation and cryopreservation.

## 5. Conclusions

Large White boars’ semen was highly susceptible to H_2_O_2_-induced oxidative stress, as shown by the compromised sperm motility and morphologicaltraits; however, antioxidants can reverse those effects. Supplementation of 5 mM dithiothreitol and 5 mM glutathione to freezing extender did not improve Large White sperm motility, viability, morphology, and acrosome integrity following cryopreservation. Supplementation of 5 mM dithiothreitol, 5 mM glutathione and a combination of 2.5 mM githiothreitol + 2.5 mM glutathione to freezing extenders did not improve fertilization of gilts oocytes. Both dithiothreitol and glutathione were not efficient in improving fertilization following in vitro fertilization. Further studies should be conducted on in vivo fertilization in order to evaluate the fertility of the frozen-thawed Large White boar semen supplemented with dithiothreitol or glutathione.

## Figures and Tables

**Figure 1 animals-12-01137-f001:**
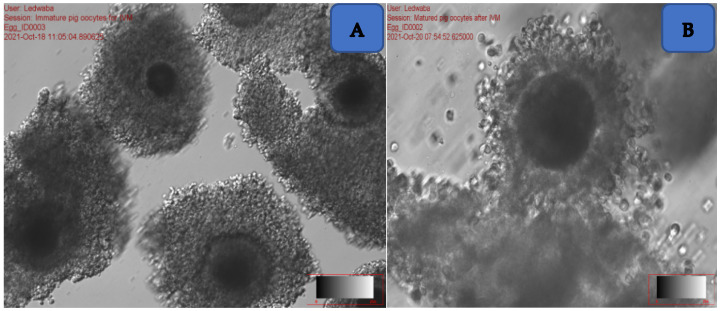
(**A**) Immature oocytes at 40× magnification; (**B**) mature oocyte at 60× magnification.

**Figure 2 animals-12-01137-f002:**
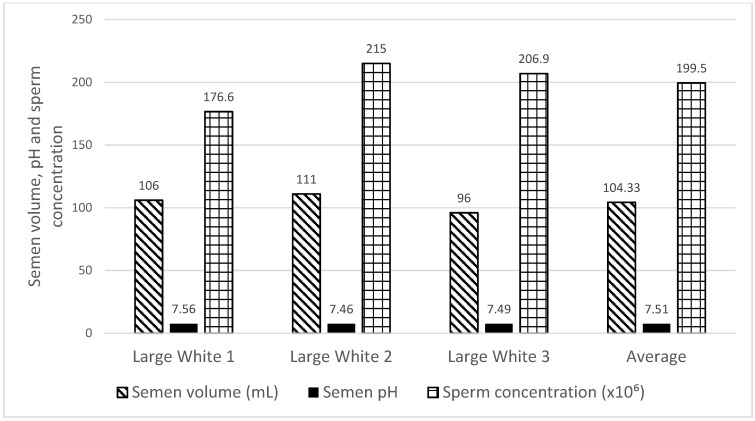
Characterization of semen volume, pH, and sperm concentration in Large White boars.

**Figure 3 animals-12-01137-f003:**
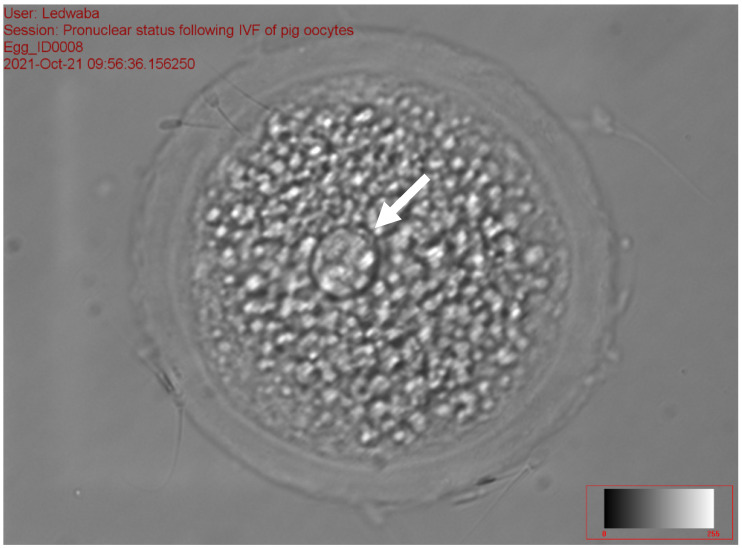
Zygote with the arrow showing 1 PN following in vitro fertilization at 40× magnification.

**Figure 4 animals-12-01137-f004:**
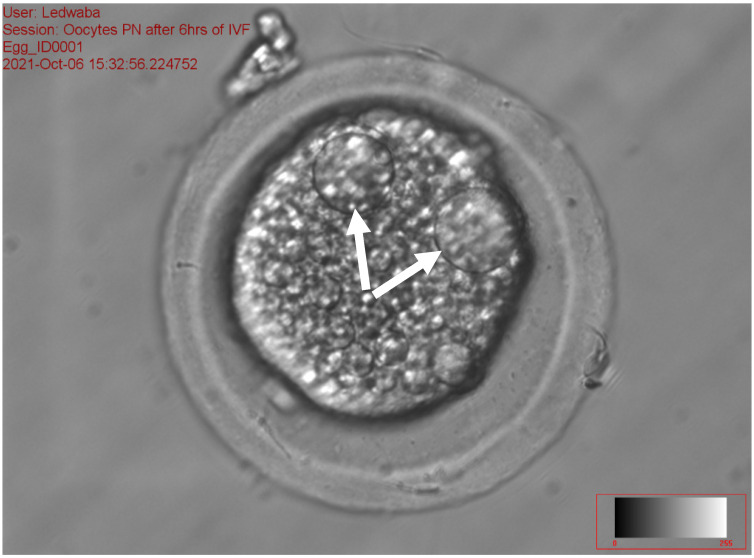
Zygote with the arrows showing 2 PN following in vitro fertilization at 40× magnification.

**Figure 5 animals-12-01137-f005:**
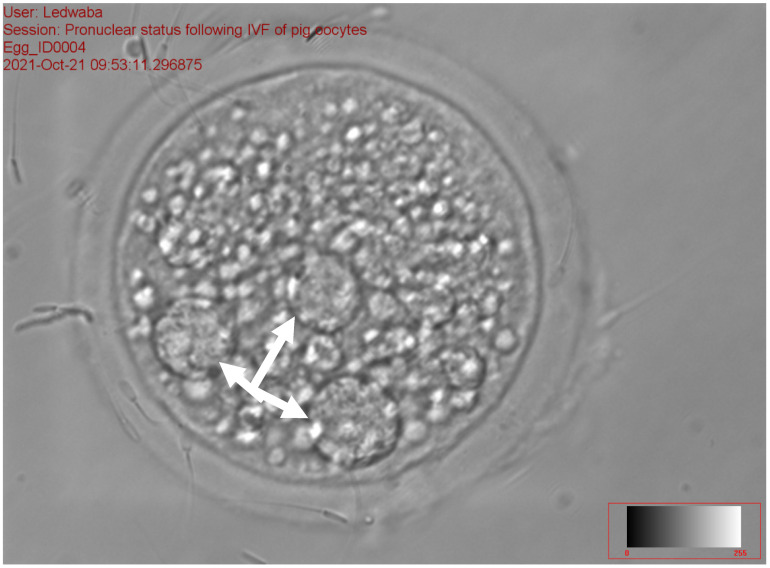
Zygote with the arrows showing >2 PN following in vitro fertilization at 40× magnification.

**Table 1 animals-12-01137-t001:** The detailed description of the CASA—Sperm Class Analyzer^®^ technology.

Parameter	Abbreviation	Definition	Unit	Reference
Total motility	TM	The ratio of motile sperm to the total cell concentration is expressed as a percentage.	%	Kathiravan et al. [21]
Progressive motility	PM	Percent of sperm moving rapidly and in a straight path.	%	Kathiravan et al. [21]
Non-progressive motility	NPM	The percentage of sperm not moving forward in a straight path.	%	Vyt et al. [22]
Static	STC	Percentage static sperm (not moving during the analysis).	%	Vyt et al. [22]
Rapid motility	RAP	Percentage of rapidly moving sperm.	%	Vyt et al. [22]
Slow	SLW	Percentage of sperm moving at 1–10 μm/second.	%	Vyt et al. [22]
Medium	MED	Percentage of sperm moving at 11–25 μm/second.	%	Vyt et al. [22]
Curvilinear velocity	VCL	The instantaneously recorded sequential progression along the whole trajectory of the sperm per unit of time.	μm/s	Somi et al. [23]
Straight-line velocity	VSL	The straight trajectory of the sperm per unit of time (=straight line distance from beginning to end of track divided by the time taken).	μm/s	Somi et al. [23]
Average path velocity	VAP	The mean trajectory of the sperm per unit of time.	μm/s	Somi et al. [23]
Linearity	LIN	The ratio of the straight displacement in the sum of elementary displacements during the time of the measurement is defined as (VSL/VCL) × 100.	%	Somi et al. [23]
Straightness	STR	The ratio of projected length to the average velocity of sperm head along a spatial trajectory, STR = VSL/VAP.	%	Somi et al. [23]
Wobble	WOB	The oscillation of the curvilinear trajectory upon the mean trajectory is defined as (VAP/VCL) × 100.	%	Somi et al. [23]
Beat-cross frequency	BCF	The number of lateral oscillatory movements of the sperm head around the mean trajectory.	Hz	Somi et al. [23]
Amplitude of lateral head displacement	ALH	The mean width of sperm head oscillation.	μm	Somi et al. [23]

**Table 2 animals-12-01137-t002:** Evaluation of sperm motility and velocity traits of fresh semen from Large White boars.

Boar Semen	TM (%)	PM (%)	NPM (%)	STC (%)	RAP (%)	MED (%)	SLW (%)	VCL (µm/s)	VSL (µm/s)	VAL (µm/s)	LIN (%)	STR (%)	WOB (%)	ALH (µm/s)	BCF (H_Z_)	HPA (%)
Fresh semen	97.93 ± 2.14	53.40 ± 10.79	44.53 ± 9.84	2.07 ± 2.14	40.63 ± 14.53	53.61 ± 13.05	2.15 ± 1.69	110.58 ± 18.25	27.46 ± 3.17	62.21 ± 9.03	27.07 ± 4.33	46.74 ± 5.50	56.45 ± 3.42	2.66 ± 0.37	26.23 ± 1.55	16.27 ± 8.38

TM = total motility; PM = progressive motility; NPM = non-progressive motility; RAP = rapid; MED = medium; SLW = slow; STC = static; VCL = curvilinear velocity; VSL = straight-line velocity; VAP = average path velocity; LIN = linearity; STR = straightness; WOB = wobble; ALH = amplitude of lateral head displacement; BCF = beat cross frequency; HPA = hyperactive.

**Table 3 animals-12-01137-t003:** Evaluation of sperm viability and morphology traits of fresh semen from Large White boars.

Boar Semen	Sperm Viability (%)		Live Sperm with Abnormalities (%)			
	Live	Dead	Head	Tail	Proximal Droplets	Distal Droplets
Fresh semen	92.80 ± 8.26	3.70 ± 4.31	0.50 ± 0.71	1.80 ± 0.92	0.70 ± 0.67	0.50 ± 0.71

**Table 4 animals-12-01137-t004:** Characterization of sperm motility and velocity traits of treated Large White boars’ semen following in vitro liquid preservation under induced oxidative stress (mean ± SD).

Antioxidants Treatments	TM (%)	PM (%)	NPM (%)	STC (%)	RAP (%)	MED (%)	SLW (%)	VCL (µm/s)	VSL (µm/s)	VAP (µm/s)	LIN (%)	STR (%)	WOB (%)	ALH (µm/s)	BCF (Hz)	HPA (%)
Control	93.32 ± 7.74	71.28 ± 17.86 ^a^	22.04 ± 15.42 ^c^	8.35 ± 7.81	64.92 ± 21.60 ^a^	22.40 ± 17.50 ^b^	6.35 ± 2.90	158.91 ± 52.38 ^a^	29.29 ± 5.39 ^ab^	87.69 ± 23.89 ^a^	20.58 ± 5.76 ^b^	35.86 ± 8.48 ^b^	55.60 ± 4.63	3.44 ± 1.07 ^a^	27.83 ± 3.11	27.71 ± 12.28
5 µM H_2_O_2_	86.70 ± 10.24	49.88 ± 16.81 ^c^	36.82 ± 16.06 ^a^	13.30 ± 10.24	40.76 ± 19.69 ^b^	38.69 ± 17.28 ^a^	6.03 ± 3.45	112.60 ± 26.79 ^b^	24.61 ± 5.16 ^b^	62.71 ± 12.50 ^b^	25.09 ± 5.68 ^ab^	43.27 ± 6.15 ^ab^	56.42 ± 6.21	2.64 ± 0.50 ^b^	26.87 ± 2.99	21.81 ± 16.37
5 µM DTT	89.72 ± 10.19	54.14 ± 12.63 ^bc^	35.58 ± 11.46 ^ab^	10.28 ± 10.19	43.48 ± 16.27 ^b^	39.70 ± 16.02 ^a^	6.55 ± 3.29	121.74 ± 22.37 ^b^	23.81 ± 3.79 ^b^	66.03 ± 10.52 ^b^	24.40 ± 8.40 ^ab^	43.03 ± 9.85 ^ab^	54.24 ± 8.36	2.84 ± 0.60 ^ab^	26.39 ± 2.41	20.47 ± 9.94
5 µM GSH	88.15 ± 9.47	63.81 ± 12.88 ^ab^	24.34 ± 13.08 ^bc^	11.85 ± 9.47	53.46 ± 17.06 ^ab^	30.26 ± 18.63 ^ab^	4.38 ± 3.52	136.57 ± 30.15 ^ab^	31.50 ± 9.46 ^a^	75.76 ± 14.31 ^ab^	24.77 ± 5.29 ^ab^	43.60 ± 8.18 ^ab^	55.68 ± 8.03	3.09 ± 0.78 ^ab^	26.91 ± 1.77	25.12 ± 12.86
5 µM H_2_O_2_ + 5 µM DTT	86.71 ± 11.22	64.70 ± 14.29 ^ab^	22.01 ± 10.85 ^c^	13.29 ± 11.22	54.44 ± 20.38 ^ab^	28.70 ± 17.03 ^ab^	4.46 ± 3.14	131.24 ± 25.61 ^ab^	29.58 ± 3.69 ^ab^	73.93 ± 8.99 ^b^	26.01 ± 8.07 ^ab^	43.14 ± 7.35 ^ab^	57.75 ± 8.56	2.95 ± 0.57 ^ab^	27.61 ± 1.55	29.74 ± 13.93
5 µM H_2_O_2_ + 5 µM GSH	90.54 ± 6.74	59.31 ± 14.32 ^abc^	27.41 ± 10.25 ^abc^	9.46 ± 6.74	42.36 ± 22.80 ^b^	38.49 ± 14.77 ^a^	6.66 ± 4.77	116.68 ± 31.31 ^b^	31.76 ± 10.32 ^a^	68.19 ± 13.19 ^b^	28.81 ± 12.91 ^a^	48.93 ± 15.09 ^a^	59.68 ± 11.42	2.67 ± 0.66 ^b^	26.14 ± 4.80	19.30 ± 13.93

^a–c^ Values with different superscripts within the column are different statistically (*p* < 0.05). TM = total motility; PM = progressive motility; NPM = non-progressive motility; RAP = rapid; MED = medium; SLW = slow; STC = static; VCL = curvilinear velocity; VSL = straight-line velocity; VAP = average path velocity; LIN = linearity; STR = straightness; WOB = wobble; ALH = amplitude of lateral head displacement; BCF = beat cross frequency; HPA = hyperactive; GSH = glutathione; DTT = dithiothreitol; H_2_O_2_ = hydrogen peroxide.

**Table 5 animals-12-01137-t005:** Characterization of sperm viability and morphology traits of treated Large White boars’ semen following in vitro liquid preservation under induced oxidative stress (mean ± SD).

Antioxidants Treatments	Sperm Viability (%)		Live Sperm with Abnormalities (%)			
	Live	Dead	Head	Tail	Proximal Droplets	Distal Droplets
Control	86.00 ± 4.22 ^a^	9.80 ± 3.99 ^bc^	0.33 ± 0.71 ^b^	2.20 ± 1.23	0.70 ± 0.82 ^b^	0.60 ± 0.52
5 µM H_2_O_2_	76.00 ± 10.46 ^b^	20.30 ± 6.78 ^a^	0.30 ± 0.48 ^b^	2.80 ± 1.62	0.50 ± 0.53 ^b^	0.10± 0.32
5 µM DTT	85.10 ± 6.51 ^a^	8.60 ± 2.59 ^c^	0.70 ± 0.95 ^ab^	3.22 ± 1.39	0.60 ± 0.70 ^b^	0.40± 0.70
5 µM GSH	77.40 ± 10.46 ^b^	14.80 ± 6.55 ^ab^	0.89 ± 0.78 ^ab^	3.89 ± 2.09	0.50 ± 0.53 ^b^	0.50± 0.70
5 µM H_2_O_2_ + 5 µM DTT	75.80 ± 10.22 ^b^	16.70 ± 6.68 ^a^	1.10 ± 1.10 ^a^	3.22 ± 2.44	1.40 ± 1.07 ^a^	0.44± 0.73
5 µM H_2_O_2_ + 5 µM GSH	79.00 ± 9.46 ^ab^	15.70 ± 9.78 ^a^	0.70 ± 0.48 ^ab^	3.60 ± 2.07	0.50 ± 0.53 ^b^	0.50± 0.53

^a–c^ Values with different superscripts within the column are different statistically (*p* < 0.05). H_2_O_2_ = hydrogen peroxide; DTT = dithiothreitol; GSH = glutathione.

**Table 6 animals-12-01137-t006:** Characterization of sperm motility and velocity traits of frozen-thawed Large White boars’ semen supplemented with antioxidants (mean ± SD).

Antioxidants Treatments	TM (%)	PM (%)	NPM (%)	STC (%)	RAP (%)	MED (%)	SLW (%)	VCL (µm/s)	VSL (µm/s)	VAP (µm/s)	LIN (%)	STR (%)	WOB (%)	ALH (µm/s)	BCF (Hz)	HPA (%)
Control	32.09 ± 6.66 ^a^	19.13 ± 6.96 ^a^	12.95 ± 4.78	67.92 ± 6.66 ^b^	17.04 ± 6.55 ^a^	8.79 ± 3.45	6.25 ± 2.89 ^ab^	111.63 ± 18.29	22.77 ± 4.66	50.74 ± 11.20	21.31 ± 3.08	44.72 ± 4.80	44.96 ± 4.33	3.19 ± 0.39	17.67 ± 5.22	18.66 ± 12.62
5 mM DTT	28.33 ± 10.66 ^ab^	16.48 ± 5.74 ^a^	13.17 ± 6.57	71.67 ± 10.66 ^ab^	13.41 ± 4.94 ^ab^	8.25 ± 3.95	6.61 ± 2.52 ^a^	114.67 ± 21.49	24.75 ± 5.36	53.59 ± 17.05	23.39 ± 3.89	47.81 ± 5.91	46.01 ± 2.32	3.19 ± 0.69	18.43 ± 5.38	21.49 ± 16.84
5 mM GSH	26.04 ± 9.41 ^ab^	15.24 ± 5.71 ^ab^	10.95 ± 5.59	73.23 ± 9.68 ^ab^	12.43 ± 6.04 ^ab^	8.20 ± 4.64	5.59 ± 2.74 ^ab^	114.72 ± 25.18	24.55 ± 4.24	53.63 ± 14.23	21.74 ± 3.67	47.11 ± 8.83	45.78 ± 4.48	3.29 ± 0.57	16.72 ± 4.09	18.25 ± 13.06
2.5 mM DTT + 2.5 mM GSH	22.45 ± 11.14 ^b^	10.41 ± 4.59 ^b^	9.49 ± 3.83	77.55 ± 11.14 ^a^	9.20 ± 3.55 ^b^	6.53 ± 2.76	4.05 ± 2.31 ^b^	123.01 ± 24.79	23.12 ± 3.92	55.43 ± 16.26	21.97 ± 3.11	47.07 ± 8.08	44.88 ± 2.96	3.69 ± 0.62	16.43 ± 5.52	20.29 ± 15.52

^a,b^ Values with different superscripts within the column are different statistically (*p* < 0.05). TM = total motility; PM = progressive motility; NPM = non-progressive motility; RAP = rapid; MED = medium; SLW = slow; STC = static; VCL = curvilinear velocity; VSL = straight-line velocity; VAP = average path velocity; LIN = linearity; STR = straightness; WOB = wobble; ALH = amplitude of lateral head displacement; BCF = beat cross frequency; HPA = hyperactive; GSH = glutathione; DTT = dithiothreitol; H_2_O_2_ = hydrogen peroxide.

**Table 7 animals-12-01137-t007:** Characterization of sperm viability, morphology, and acrosome integrity of frozen-thawed Large White boars’ semen supplemented with antioxidants (mean ± SD).

Antioxidants Treatments	Acrosome Integrity (%)		Viability (%)		Live Sperm Abnormalities (%)			
	Reacted Acrosome	Non-Reacted Acrosome	Live	Dead	Head Defects	Tail Defects	Proximal Droplets	Distal Droplet
Control	25.20 ± 5.39	74.80 ± 5.39	34.90 ± 6.51 ^a^	62.50 ± 6.42 ^b^	0.33 ± 0.50	1.22 ± 0.67	0.20 ± 0.42	0.30 ± 0.48
5 mM DTT	22.40 ± 13.16	77.60 ± 13.16	29.80 ± 5.20 ^a^	67.40 ± 5.78 ^b^	0.20 ± 0.42	1.70 ± 0.82	0.50 ± 0.53	0.40 ± 0.52
5 mM GSH	24.40 ± 11.65	75.60 ± 11.65	29.40 ± 6.38 ^a^	68.40 ± 6.79 ^b^	0.50 ± 0.53	1.10 ± 0.74	0.30 ± 0.48	0.30 ± 0.48
2.5 mM DTT + 2.5 mM GSH	24.50 ± 14.14	75.50 ± 14.14	21.67 ± 6.91 ^b^	76.00 ± 7.48 ^a^	0.30 ± 0.48	1.70 ± 0.67	0.33 ± 0.50	0.10 ± 0.32

^a,b^ Values with different superscripts within the column are different statistically (*p* < 0.05). DTT = dithiothreitol; GSH = glutathione.

**Table 8 animals-12-01137-t008:** The effect of antioxidants on sperm fertilization ability following in vitro maturation of gilts oocytes in vitro fertilization (mean ± SD).

Antioxidants Treatments	IVM Oocytes (*n*)	IVF Oocytes (*n*)	Pronucleus				Total Number of Fertilization Rate (%)	Total Number of Non-Fertilization Rate (%)
			0 PN *n* (%)	1 PN *n* (%)	2 PN *n* (%)	>2 PN *n* (%)		
Fresh semen	125	97	53.96 ± 11.83	22.50 ± 7.13	10.02 ± 8.76	11.84 ± 9.47 ^a^	46.04 ± 11.83	53.96 ± 11.83
Control	129	80	51.52 ± 17.02	28.12 ± 9.34	13.32 ± 7.22	7.04 ± 5.10 ^ab^	48.48 ± 17.02	51.52 ± 17.02
5 mM GSH	116	82	68.06 ± 8.65	18.32 ± 3.92	11.18 ± 5.06	0.86 ± 1.92 ^b^	31.94 ± 8.65	68.06 ± 8.65
5 mM DTT	135	114	55.26 ± 15.75	22.76 ± 17.93	19.76 ± 9.90	2.22 ± 4.96 ^b^	44.74 ± 15.75	55.26 ± 15.75
2.5 mM DTT + 2.5 mM GSH	120	79	51.28 ± 16.07	22.80 ± 9.55	11.82 ± 11.57	14.10 ± 10.49 ^a^	48.72 ± 16.07	51.28 ± 16.07

^a,b^ Values with different superscripts within the column are different statistically (*p* < 0.05). DTT = dithiothreitol; GSH = glutathione; PN = pronucleus; IVM = in vitro maturation; IVF = in vitro fertilization.

**Table 9 animals-12-01137-t009:** Pearson correlation coefficient for boar sperm traits (sperm motility and velocity traits) and the oocyte fertilization (pronucleus) rate.

Traits	0 PN %	1 PN %	2 PN %	<2 PN %	FR %	TM %	PM %	NPM %	STC %	RAP %	MED %	SLW %	VCL µm/s	VSL µm/s	VAP µm/s	LIN %	STR %	WOB %
0 PN%	1.00																	
1 PN%	−0.59	1.00																
2 PN%	−0.56	−0.02	1.00															
<2 PN%	−0.43	−0.18	−0.06	1.00														
FR%	−1.00	0.59	0.56	0.43	1.00													
TM%	0.06	−0.09	−0.22	0.23	−0.06	1.00												
PM%	0.11	−0.18	−0.23	0.26	−0.11	0.97	1.00											
NPM%	−0.01	0.05	−0.02	−0.02	0.01	0.58	0.47	1.00										
STC%	−0.06	0.08	0.15	−0.14	0.06	−0.90	−0.87	−0.86	1.00									
RAP%	0.12	−0.20	−0.25	0.28	−0.12	0.92	0.98	0.38	−0.79	1.00								
MED%	−0.04	0.01	−0.18	0.25	0.04	0.96	0.88	0.62	−0.87	0.81	1.00							
SLW%	0.04	0.04	0.14	−0.26	−0.04	−0.12	−0.16	0.72	−0.32	−0.25	−0.09	1.00						
VCL µm/s	0.15	−0.21	−0.17	0.17	−0.15	−0.05	0.14	−0.23	0.05	0.25	−0.17	−0.17	1.00					
VSL µm/s	0.13	−0.25	0.01	0.06	−0.13	−0.03	0.08	−0.20	0.07	0.12	−0.09	−0.17	0.48	1.00				
VAP µm/s	0.24	−0.31	−0.30	0.27	−0.24	0.15	0.32	−0.24	−0.05	0.41	0.06	−0.38	0.90	0.42	1.00			
LIN%	0.19	−0.11	−0.12	−0.06	−0.19	0.16	0.05	0.07	−0.07	−0.04	0.23	−0.06	−0.54	0.35	−0.37	1.00		
STR%	−0.02	0.001	0.22	−0.21	0.02	−0.11	−0.21	0.10	0.07	−0.28	−0.05	0.21	−0.49	0.45	−0.57	0.79	1.00	
WOB%	0.23	−0.18	−0.50	0.34	−0.23	0.51	0.49	0.01	−0.29	0.46	0.57	−0.47	−0.01	0.02	0.39	0.40	−0.20	1.00

TM = total motility; PM = progressive motility; NPM = non-progressive motility; RAP = rapid; STC = static; VCL = curvilinear velocity; VSL = straight-line velocity; VAP = average path velocity; LIN = linearity; STR = straightness; WOB = wobble; PN = pronucleus; FR = fertilization rate.

## Data Availability

The data are available from the corresponding author.

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
