# Peer review of "Investigation of the Efficacy of Dithiothreitol and Glutathione on In Vitro Fertilization of Cryopreserved Large White Boar Semen"

_animals, 2022, doi:10.3390/ani12091137_

Round 1

Reviewer 1 Report

"Investigation of the efficacy of Dithiothreitol and Glutathione on in vitro fertilization of cryopreserved Large white boars semen" is an article based on the evaluation of the antioxidant activity of Dithiothreitol (DTT) and Glutathione (GSH) on the sperm of Large white boars subjected to stress oxidative and cryopreserved.

Basically the work is interesting and the results could create a new step in the experimentation of substances with antioxidant action on animal semen. Unfortunately, the work has various gaps, which I will try to report below.

In the introduction, L. 63, after stating that "boar sperm cells are rich in membrane polyunsaturated fatty acid" it would be appropriate to add a reference.

The introduction delves a lot on the action of ROS, rightly, but the part relating to the bibliographic bases that led to the decision to use GSH and DTT as antioxidants is missing. In fact the references no. 15 and 16, which should explain these reasons, deal with antioxidants such as L-carnitine, selenium, vitamin C and vitamin E in humans and cryopreserved bull semen, respectively. So I recommend revising L. 79-80. As a consequence of this observation, I would like to ask you what were the reasons, I hope reinforced by bibliographic evidence, that made you decide for the doses of GSH and DTT used for the present study.

The part concerning Materials and Methods, in my opinion, should be made more orderly and clearer.

The first section should be completed with other information, concerning for example the duration of the study; in what time frame the study was carried out; if the animals have been handled in accordance with the European Commission Directive for pig welfare.

I would add, in this paragraph, the experimental design with an indication of: how many animals were used, when the semen was collected and how often; what are the study groups; fresh and cryopreserved semen evaluation, with the respective dosages; the characterization of semen under oxidative stress; how the seed is evaluated (with description of the CASA-system) and description of the contents of Table 1, remembering to write “Table 1” in the text. In this regard, I would recommend: redo the numbering of the Figures (there are two Figures with the number 1) and insert them immediately after having cited them in the text, in an adequate and graphically appropriate position. This also applies to the Tables.

“Ethical clearance” could be moved at the end of the work before the References.

“Treatments and mediums” paragraph could be included in “In vitro maturation of oocytes”.

Results: there are tables and figures which, however, are not indicated in the text. Furthermore, in the discussion there is a rather substantial part that talks about polyspermia, but there is no correspondence in the "Results":

Format the bibliography according to the rules of the newspaper.

Review the English language carefully.

Author Response

RESPONSES TO REVIEWER COMMENTS

Title: “Investigation of the efficacy of Dithiothreitol and Glutathione on in vitro fertilization of cryopreserved Large white boars semen

Re:      Ms. Ref. No.:  Animal 1643930                                            Ledwaba et al.

Responses to Reviewer No 1

Our specific response to your comments follows.

Comment: In the introduction, L. 63, after stating that "boar sperm cells are rich in membrane polyunsaturated fatty acid" it would be appropriate to add a reference.

Response: Corrected. The reference was added on the referred sentence and it reads as follows:

  • “Boar sperm cells, specifically, are rich in membrane polyunsaturated fatty acids [5] which are a target for ROS”.
  • Reference: Brouwers, J.F.; Silva, P.F.N.; Gadella, B.M. New assay for detection and localization of endogenous lipid peroxidation products in living boar sperm after BTS dilution or after freeze-thawing. Theriogenology. 2005, 63, 458-469.

Comment: The introduction delves a lot on the action of ROS, rightly, but the part relating to the bibliographic bases that led to the decision to use GSH and DTT as antioxidants is missing. In fact the references no. 15 and 16, which should explain these reasons, deal with antioxidants such as L-carnitine, selenium, vitamin C and vitamin E in humans and cryopreserved bull semen, respectively. So I recommend revising L. 79-80. As a consequence of this observation, I would like to ask you what were the reasons, I hope reinforced by bibliographic evidence, that made you decide for the doses of GSH and DTT used for the present study.

Response: Corrected. More information which support the decision to use GSH and DTT was provided.

  • “Addition of GSH to the freezing and thawing extender would be expected to improve the quality and fertilizing ability of frozen-thawed boar sperm [16; 17], since the addition of GSH helps to maintain sperm motility [18; 19] and to protect sperm against oxidative damage [20]. Dithiothreitol is known as an antioxidant and it decreases protamine disulfide bond [21]. It provides a protective effect against apoptosis and oxidative damage [22]. Other studies have indicated that dithiothreitol (DTT) can prevent hydrogen peroxide (H2O2) meditated loss of boar semen quality [23; 24]. The addition of DTT seems to improve bull and human sperm motility during liquid storage or in the frozen state. However, this was never tested on the boar semen”.

Comment: The part concerning Materials and Methods, in my opinion, should be made more orderly and clearer.

Response: Corrected. The materials and method part was explained and made orderly and clearer.

Comment: The first section should be completed with other information, concerning for example the duration of the study; in what time frame the study was carried out; if the animals have been handled in accordance with the European Commission Directive for pig welfare.

Response: Corrected. The first section of materials and methods was completed with more information which covers the duration of the study, time frame and handling of the animals.

  • “Three Large White boars (approximately 3 years old) were used for the study. These animals were raised and trained on the farm as semen donors for AI purposes. They were maintained under uniform feeding and housing conditions. Animals were fed once daily and water was provided ad libitum. The study was carried out for the duration of three months. Ovaries were collected from the prepubescent gilts of un-known breeds from the local abattoir. A completely randomized design was used for this study”.

Comment: I would add, in this paragraph, the experimental design with an indication of: how many animals were used, when the semen was collected and how often; what are the study groups; fresh and cryopreserved semen evaluation, with the respective dosages; the characterization of semen under oxidative stress; how the seed is evaluated (with description of the CASA-system) and description of the contents of Table 1, remembering to write “Table 1” in the text. In this regard, I would recommend: redo the numbering of the Figures (there are two Figures with the number 1) and insert them immediately after having cited them in the text, in an adequate and graphically appropriate position. This also applies to the Tables.

Response: Corrected. The experimental design was explained and clarified under the ‘materials’ section. The evaluation of fresh and cryopreserved semen was also clarified and explained in details. The numbering of the table and figures have been revised and labelled accordingly.

  • “After 3 hours of incubation, a drop of 5 µL of semen was placed on a pre-warmed microscope slide (Labchem Pty Ltd, South Africa) and covered with a cover slip (Labchem Pty Ltd, South Africa), then examined under a microscope using Sperm Class Analyser. The sperm motility and velocity traits measured (Table 1) includes the sperm total motility (TM %), progressive motility (PM %), non-progressive motility (NPM %), static (STC %), rapid (RAP %), medium (MED %), slow (SLW %), curvilinear (VCL µm/s), straight line (VSL µm/s), average path velocity (VAP µm/s), linearity (LIN %), straightness (STR %), wobble (WOB %), amplitude of lateral head displacement (ALH µm/s), beat cross frequency (BCF Hz) and hyperactivity (HPA %). The sperm viability and morphology traits were evaluated using the Eosin/ Nigrosin (University of Pretoria, Faculty of Veterinary Science Pharmacy, Onderstepoort, South Africa) staining. Briefly, a drop of 7 µL of the semen was added into the 20 µL of Eosin/Nigrosin staining. A drop of 5 μL mixed sample was placed on the end of microscope slide and smeared to the other end of the microscope slide. Thereafter the slide was left to dry off at room temperature for 5-10 minutes before evaluation. Fluorescent microscope (Olympus Corporation BX 51FT, Tokyo, Japan) was used at 100x magnification to count 200 sperms per each stained slide. For this analysis, sperm viability (live and dead) was recorded and sperm abnormalities (live sperms with head defects, live sperms with tail defects, live sperms with midpiece defects and live sperms with drop-lets defects) were recorded”.

Comment: “Ethical clearance” could be moved at the end of the work before the References.

Response: Corrected. The ethical clearance have been moved at the end of the work, before the reference.

Comment: “Treatments and mediums” paragraph could be included in “In vitro maturation of oocytes”.

Response: Corrected. The treatments and mediums paragraph have been moved and included on the In vitro maturation of oocytes paragraph.

  • 5.2. In Vitro Maturation of Oocytes

“The base maturation medium consisted of North California States University 23 (NCSU 23) supplemented with 10 ng/mL of follicle stimulating hormone, 10 ng/mL of Luteinizing hormone and 10 % porcine follicular fluid (pFF) recovered from prepubertal follicles. The fertilization medium consisted of modified Tris buffered medium (mTBM) containing 113.1 mM Sodium chloride, 3 mM Potassium chloride, 7.5 mM Calcium chloride dihydrate, 20 mM Tris, 11 mM Glucose, 5 mM Na-Pyruvate, 1 mM Caffeine and 0.1 % Bovine serum albumin (BSA). For staining of the oocytes, 25 mg (0.025 g) of Hoechst 33342 (Sigma B226) was pre-pared in 2.5 mL of pure water to make stock A. The concentration of the solution was 10 mg/mL. On the day of use, 10 μL of stock A was diluted in 10 mL of Dulbecco's phosphate buffered saline (DPBS) supplemented with 20 % (2 mL) Glycerol to make stock B (8 mL of DPBS and 2 mL of glycerol). The final concentration for stock B was 10 μg/mL”.

Comment: Results: there are tables and figures which, however, are not indicated in the text. Furthermore, in the discussion there is a rather substantial part that talks about polyspermia, but there is no correspondence in the "Results":

Response: Corrected. Both tables and figures have been double checked and included in the text. In the results, the polyspermia is represented by the “>2 PN”.

  • “The macroscopic evaluations of the Large White boars’ results are set out on the figure 2”.
  • “Raw sperm TM (97.93±2.14) and RAP (40.63±14.53) was recorded (Table 1). Raw semen obtained from Large White boars showed 92.80 % of viable sperm morphology (Table 2)”.
  • “The properties of sperm motility and velocity traits following in vitro liquid preservation under induced oxidative stress following incubation for 3 hours are evaluated (Table 3)”.
  • “Sperm morphology of above 70 % was recorded from Large White boars’ raw semen (Table 4)”.
  • “The average sperm TM for frozen thawed semen was 27.23 % for all treatments (Table 5)”.
  • “The average live sperm of frozen thawed semen was 39.71 % for all treatments (Table 6)”.
  • “The results of the effect of antioxidants on sperm fertilization ability following IVM of pig oocytes in vitro are presented in Table 7”.
  • “The Pearson’s correlation coefficients between sperm parameters results evaluated by the computer assisted sperm analysis and fertilization rate of pig oocytes are set out in table 8”.

Comment: Format the bibliography according to the rules of the newspaper.

Response: Corrected. The bibliography have been checked and formatted accordingly.

Responses to general comments:

Thank you very much for your careful and critical reading of our manuscript. All your comments are very helpful and we have revised our manuscript accordingly. We believe the mansucript has now been improved significantly.

Reviewer 2 Report

     1,  In 2.5. Characterization of Large White boars’ semen under oxidative stress, there are six treatments:control, 5 µM H2O2, 5µM DTT, 5µM GSH, 5µM H2O2+5µM DTT, 5 µM H2O2+5 µM GSH. How did you select 5µM concentration instead of 1, 3, 7, 10µM and so on. You should do something in your design or some explanation for their concerntration of selection.  Similar explanation should be provided about in 2.6. Cryopreservation of Large White boars’ semen (four treatments (control, 5 mM GSH, 5 mM 157 DTT, and combination of 2.5 mM GSH + 2.5 mM DTT).

    2,  Two pictures in Figure 1. show that the maturation cultural time is 43h, not 44h. also two picture A and B are not of same magnification.

Author Response

RESPONSES TO REVIEWER COMMENTS

Title: “Investigation of the efficacy of Dithiothreitol and Glutathione on in vitro fertilization of cryopreserved Large white boars semen

Re:      Ms. Ref. No.:  Animal 1643930                                            Ledwaba et al.

Responses to Reviewer No 2

Our specific response to your comments follows.

Comment: In 2.5. Characterization of Large White boars’ semen under oxidative stress, there are six treatments:control, 5 µM H2O2, 5µM DTT, 5µM GSH, 5µM H2O2+5µM DTT, 5 µM H2O2+5 µM GSH. How did you select 5µM concentration instead of 1, 3, 7, 10µM and so on. You should do something in your design or some explanation for their concerntration of selection.  Similar explanation should be provided about in 2.6. Cryopreservation of Large White boars’ semen (four treatments (control, 5 mM GSH, 5 mM 157 DTT, and combination of 2.5 mM GSH + 2.5 mM DTT).

Response: The preliminary study was done in order to select the concentrations for both objectives. Moreover, the concentrations were also selected based on the support from the literature study that was done prior data collection. Explanation with regards to the concentration have been provided under ‘materials and methods” section.

  • “Experiment I covered the characterization of boar semen under oxidative stress, a total number of 30 ejaculates (10 ejaculates from each Large White boar) were collected from three Large White boars twice a week for the duration of 10 weeks. The preliminary study was done in order to select suitable concentration (1 µM, 2 µM, 3 µM, 4 µM and 5 µM) for H2O2, DTT and GSH for liquid preservation.”
  • “Experiment II covered boar semen cryopreservation and evaluation, a total number of 30 ejaculates (10 ejaculates from each Large White boar) were collected from three Large White boars twice a week for the duration of 10 weeks. The preliminary study was done in order to select suitable concentration (2.5 mM, 5 mM, 7.5 mM and 10 mM) for DTT and GSH for cryopreservation of Large White boar semen. The concentration was also supported by literature review”.
  • “Experiment III covered the evaluation of the ability of cryopreserved Large White boar semen to fertilize the pig oocytes. A total of 250 ovaries were collected (50 ovaries per treatment) were collected for 5 days a week for the duration of 5 weeks. For this experiment, four treatments (fresh semen, control, 5 mM DTT, 5 mM GSH and combination of 2.5 mM DTT and 2.5 mM GSH) from the cryopreserved semen were used in or-der to evaluate their ability to fertilize pig oocytes”.

Comment: Two pictures in Figure 1. show that the maturation cultural time is 43h, not 44h. also two picture A and B are not of same magnification.

Response: Corrected. The protocol used for maturation cultural time was 44h. The magnifications were labelled properly for each picture.

  • Figure 1. A= Immature oocytes at 40X magnification and; B= Matured oocyte at 60X magnification.

Responses to general comments:

Thank you very much for your careful and critical reading of our manuscript. All your comments are very helpful and we have revised our manuscript accordingly. We believe the manuscript has now been improved significantly.

Round 2

Reviewer 1 Report

The authors answered most of the suggestions and comments comprehensively.

A few points remain to be clarified, listed below:

2.1. paragraph should be entitled: "Animals and Materials".

L. 140: Please, put the brand and origin of the "Sperm Class Analyzer", as you did for the microscope.

L. 164: Please, provide references.

Figures 1b, Figures 3, 4 and 5 are not mentioned in the text.

Figures 3, 4, 5 are inserted at the end of the results, but there is no specific paragraph describing them. As already reported in the previous comments, the discussion paragraph comments on something not reported in the resuls. So please, add a new subparagraph and cite the figures.

Author Response

RESPONSES TO REVIEWER COMMENTS

Title: “Investigation of the efficacy of Dithiothreitol and Glutathione on in vitro fertilization of cryopreserved Large white boars semen

Re:      Ms. Ref. No.: Animal 1643930                                             Ledwaba et al.

Responses to Reviewer No 1 (Round 2)

Our specific response to your comments follows.

Comment: 2.1. Paragraph should be entitled: "Animals and Materials".

Response: Corrected. The paragraph for 2.1 have been corrected and it reads as follows:

  • “2.1. Animals and Materials”.

Comment: L. 140: Please, put the brand and origin of the "Sperm Class Analyzer", as you did for the microscope.

Response: Corrected. The brand and origin of the Sperm Class Analyzer was included.

  • “After 3 hours of incubation, a drop of 5 µL of semen was placed on a pre-warmed microscope slide (Labchem Pty Ltd, South Africa) and covered with a cover slip (Labchem Pty Ltd, South Africa), then examined under a microscope using Sperm Class Analyser (Microptic S.L, Barcelona)”.

Comment: L. 164: Please, provide references.

Response: Corrected. The references have been provided.

Parameter

Abbreviation

Definition

Unit

Reference

Total motility

TM

The ratio of motile cells to the total cell concentration expressed as percentage.

%

Kathiravan et al. [26]

Progressive motility

PM

Percent of sperm moving rapidly and in a straight path

%

Kathiravan et al. [26]

Non progressive motility

NPM

The percentage of sperm not moving forward in a

straight path.

%

Vyt et al.[27]

Static

STC

Percentage static spermatozoa (not moving during the analysis)

%

Vyt et al. [27]

Rapid motility

RAP

Percentage rapidly moving spermatozoa.

%

Vyt et al. [27]

Slow

SLW

Percentage of sperm moving at 1-10 μm/second.

%

Vyt et al. [27]

Medium

MED

Percentage of sperm moving at 11-25 μm/second.

%

Vyt et al. [27]

Curvilinear velocity

VCL

The instantaneously recorded sequential progression along the whole trajectory of the spermatozoon per unit of time.

μm/s

Somi et al. [28]

Straight Line velocity

VSL

The straight trajectory of the spermatozoa per unit of time (= straight line distance from beginning to end of track divided by time taken).

μm/s

Somi et al. [28]

Average path velocity

VAP

The mean trajectory of the spermatozoa per unit of time.

μm/s

Somi et al. [28]

Linearity

LIN

The ratio of the straight displacement in the sum of elementary displacements during the time of the measurement and it is defined as (VSL/VCL) x 100.

%

Somi et al. [28]

Straightness

STR

Ratio of projected length to average velocity of sperm head along a spatial trajectory, STR = VSL/VAP.

%

Somi et al. [28]

Wobble

WOB

Which indicates the oscillation of the curvilinear trajectory upon the mean trajectory and is defined as (VAP/VCL) x 100.

%

Somi et al. [28]

Beat-cross frequency

ACF

The number of lateral oscillatory movements of the sperm head around the mean trajectory.

Hz

Somi et al. [28]

Amplitude of lateral head displacement

ALH

Which is the mean width of sperm head oscillation.

μm

Somi et al. [28]

Comment: Figures 1b, Figures 3, 4 and 5 are not mentioned in the text.

Response: Corrected. The Figures have been mentioned on the texts. Figure 1b is mentioned under “In vitro maturation of oocytes”. Figure 3,4 and 5 are mentioned under “the results”.

  • “After 44 hours of incubation, matured oocytes (Figure 1b) were identified by the presence of expanded cumulus cells and were used for IVF using fresh and frozen thawed Large White semen”.
  • “The raw semen (11.84±9.47) and the combination of 2.5 mM DTT + 2.5 mM GSH (14.10±10.49) recorded high percentage of zygotes with >2 PN (Figure 5) as compared to the 5 mM GSH (0.86±1.92) and 5 mM DTT treatments (2.22±4.96) respectively (P < 0.05)”.
  • “The control treatment (28.1%) recorded high percentage of zygotes with 1 PN (Figure 3) while the 5 mM DTT treatment (19.8%) recorded high percentage of zygotes with 2 PN (figure 4), (P > 0.05)”.

Comment: Figure 3, 4, 5 are inserted at the end of the results, but there is no specific paragraph describing them. As already reported in the previous comments, the discussion paragraph comments on something not reported in the results. So please, add a new subparagraph and cite the Figures.

Response: Corrected. The paragraph describing the Figures is included under the “results”.

  • “Figure 3, 4 and 5 represent the results for in vitro fertilization of pig oocytes. While Figure 3 represent a 1 PN, which indicates that fertilization occurred but only one gamete produced a pronuclear structure but the DNA from one gamete (the sperm) is missing and only 1 PN (female pronuclei) was present. Therefore, the average for zygote with 1 PN for all treatments was 22.9 % (Table 7). Figure 4 represent normal fertilization whereby the zygote had 2 PN (female and male pronuclei). This Figure indicate that only one sperm penetrated the matured oocyte and was able to fertilize the matured oocyte. Therefore, the fertilization was successful and the zygote can reach the in vitro culture stage and develop into an embryo. The average percentage for zygotes with normal fertilization (2 PN) was 13.22 % for all the treatments (Table 7). Figure 5 represent polyspermy whereby more than one sperm fertilized matured oocyte. Even though fertilization occurred, survival of the zygote to embryo stage is low as there is uneven number of chromosomes due to extra pronuclei. Therefore, the zygote with 3 pronucleus states tend to develop into poorer-quality embryos. The average percentage for polyspermy (>2 PN) for all the treatments was 7.21 % (Table 7)”.

Responses to general comments:

Thank you very much for your careful and critical reading of our manuscript. All your comments are very helpful and we have revised our manuscript accordingly. We believe the manuscript has now been improved significantly.
